# An Adaptive Derivative Estimator for Fault-Detection Using a Dynamic System with a Suboptimal Parameter

**Manuel Schimmack [1,2] and Paolo Mercorelli [1,*]** 

[1] Institute of Product- and Processinnovation, Leuphana University of Lueneburg, Volgershall 1, D-21339 Lueneburg, Germany; schimmack@uni-leuphana.de or mas@tf.uni-kiel.de

[2] Institute for Electrical Engineering and Information Technology, Kiel University, Kaiserstraße 2, D-24143 Kiel, Germany

[*] Correspondence: mercorelli@uni-leuphana.de; Tel.: +49-4131-677-5571; Fax: +49-4131-677-5300

**Abstract:** This paper deals with an approximation of a first derivative of a signal using a dynamic system of the first order. After formulating the problem, a proposition and a theorem are proven for a possible approximation structure, which consists of a dynamic system. In particular, a proposition based on a Lyapunov approach is proven to show the convergence of the approximation. The proven theorem is a constructive one and shows directly the suboptimality condition in the presence of noise. Based on these two results, an adaptive algorithm is conceived to calculate the derivative of a signal with convergence in infinite time. Results are compared with an approximation of the derivative using an adaptive Kalman filter (KF).

**Keywords:** derivative approximation; estimator; Lyapunov approach; least squares method

## 1. Introduction

The derivative estimation of a measured signal has considerable importance in signal processing, numerical analysis, control engineering, and failure diagnostics, among others [1]. Derivatives and structures using derivatives of signals are often used in industrial applications, for example PD controllers. These kinds of controllers are often used practically in different fields of application.

In applications, signals are corrupted by measurement and process noise, therefore a filtering procedure needs to be implemented. A number of different approaches have been proposed based on least-squares polynomial fitting or interpolation for off-line applications [1,2]. Another common approach is based on high-gain observers [3–5]. These observers adjust the model by weighting the observer output deviations from the output of the system to be controlled.

In [6], a sliding mode control (SMC) using an extended Kalman filter (EKF) as an observer for stimulus-responsive polymer fibers as soft actuator was proposed. Because of the slow velocity of the fiber, the EKF produces poor estimation results. Therefore, a derivative approximation structure is proposed to estimate the velocity through the measurement of the position. This approach realized the approximation of the derivative using also a high gain observer. The method presented in the different applications cited above is a method that approximates the derivative in an infinite horizon of time. In this sense, the proposed differentiator is an asymptotic estimator of the derivative. Several researchers studied this problem by applying the SMC approach.

This paper emphasizes some mathematical aspects of an algorithm that was used in the past for practical applications such as, for instance, in [7,8]. In particular, in [7] this algorithm is used in designing a velocity observer. In [8], a similar algorithm is used to estimate parameter identification in an application in which a synchronous motor is proposed.

In recent years, real-time robust exact differentiation has become the main problem of output-feedback high order of sliding mode (HOSM) control design. Even the most modern differentiators [9] do not provide for exact differentiation with finite-time convergence and without considering noise.

The derivatives may be calculated by successive implementation of a robust exact first-order differentiator [10] with finite-time convergence but without considering noise, as in [11,12]. In [13], an arbitrary-order finite-time-convergent exact robust differentiator is constructed based on the HOSM technique.

In [10], the proposed differentiator provides the proportionality of the maximal differentiation error to the square root of the maximal deviation of the measured input signal from the base signal. Such an order of the differentiation error is shown to be the best possible when the only information known of the base signal is an upper bound for Lipschitz's constant of the derivative. According to Theorem 2, the proposed algorithm to produce the optimal approximation needs to work the knowledge of the maximal Lipschitz's constant. More recently in [14], sliding mode (SM)-based differentiation is shown to be exact for a large class of functions and robust with respect to noise. In this case, Lipschitz's constant must be known to apply the algorithm. Those methods, using the maximal Lipschitz's constant, perform an approximation in a finite horizon of time. In practical applications, the presence of noise and faults does not allow a Lipschitz's constant to be set. In fact, the noise is not distinguishable from the input signal. In this sense, it appears impossible to apply these algorithms in real applications.

This paper proposes an approximated derivative structure to be taken into account for such types of applications, so that spikes, noise, and any other kind of undesired signals that occur from the derivatives can be reduced. Thus, the problem of the approximation of the derivative is formulated in the presence of white Gaussian noise and in a infinity horizon of time as in KF. Therefore, the comparison is shown just with the performances of the approximation of the derivative performed by an adaptive KF. After the problem formulation, this paper proves a proposition which allows us to build this possible approximation of the derivative using a dynamic system.

The paper is structured as follows. In Section 2, the problem formulation and a possible solution are proposed. How to approximate a derivative controller using an adaptive KF is presented in Section 3. The results of the simulations are discussed in Section 4, and the conclusion closes the paper.

## 2. An Approximated Derivative Structure

Using the derivative structures, imprecision occurs. The imprecision is due to spikes generating power dissipation. The idea is to find an approximated structure of general derivatives as they occur in mathematical calculations, and which are often used also in technical problems as proportional derivative controllers. The following formulation states the problem in a mathematical way.

**Problem 1.** *Assume the following differential is given:*

$$r(t) = \frac{\mathrm{d}y(t)}{\mathrm{d}t}. \tag{1}$$

*Function $y(t) \in \mathbb{R}$ is the function to be differentiated, where $t \in \mathbb{R}$ represents the time variable. The aim of the proposed approach is to look for an approximating expression $\hat{r}(t) = \hat{r}\left(y(t), k_{\mathrm{app}}\right)$, where $k_{\mathrm{app}}$ is a parameter, such that:*

$$\lim_{k_{\mathrm{app}} \to +\infty} e_r(t) = r(t) - \hat{r}(t) = 0, \tag{2}$$

*where $r(t)$ represents the real derivative function.*

**Proposition 1.** *Considering (1), then there exists a function $\mathcal{M} > 0$ such that if the following dynamic system is considered:*

$$\frac{d\hat{r}(t)}{dt} = -\mathcal{M}\left(\hat{r}(t) - \frac{dy(t)}{dt}\right),\tag{3}$$

*where $y(t)$ represents a twice differentiable real function and $\hat{r}(t)$ the approximated derivative function. If*

$$\frac{d\hat{r}(t)}{dt} > \frac{dr(t)}{dt}\quad \forall t,\tag{4}$$

*and*

$$\mathcal{M} > 0\ \ \forall t,\tag{5}$$

*then*

$$\lim_{t \to +\infty} e_r(t) = 0.\tag{6}$$

**Proof.** Considering the following approximate dynamic system:

$$\frac{d\hat{r}(t)}{dt} = -\mathcal{M}\left(\hat{r}(t) - \frac{dy(t)}{dt}\right),\tag{7}$$

where $\mathcal{M}$ can be a function of $y(t)$ or a parameter with $\mathcal{M} \in \mathbb{R}$, if

$$e_r(t) = r(t) - \hat{r}(t),\tag{8}$$

then

$$\frac{de_r(t)}{dt} = \frac{dr(t)}{dt} - \frac{d\hat{r}(t)}{dt}.\tag{9}$$

After inserting (7) into (9), it follows that

$$\frac{de_r(t)}{dt} = \frac{dr(t)}{dt} + \mathcal{M}\left(\hat{r}(t) - \frac{dy(t)}{dt}\right),\tag{10}$$

if (1) is taken into consideration, then (10) becomes as follows:

$$\frac{de_r(t)}{dt} + \mathcal{M}e_r(t) = \frac{dr(t)}{dt}.\tag{11}$$

If the following Lyapunov function is considered:

$$V\left(e_r(t)\right) = \frac{1}{2}e_r^2(t),\tag{12}$$

and considering that:

$$\frac{dV\left(e_r(t)\right)}{dt} = e_r(t)\frac{de_r(t)}{dt},\tag{13}$$

according to (11), it is possible to write the following expression:

$$e_r(t) = \frac{\frac{dr(t)}{dt} - \frac{de_r(t)}{dt}}{\mathcal{M}},\tag{14}$$

and thus from (13), it follows that:

$$\frac{dV\left(e_r(t)\right)}{dt} = \frac{\frac{de_r(t)}{dt}\frac{dr(t)}{dt} - \left(\frac{de_r(t)}{dt}\right)^2}{\mathcal{M}}.\tag{15}$$

Considering (9) and multiplying by $\frac{de_r(t)}{dt}$, then

$$\frac{de_r(t)}{dt}\frac{dr(t)}{dt} = \left(\frac{dr(t)}{dt}\right)^2 - \frac{d\hat{r}(t)}{dt}\frac{dr(t)}{dt}, \tag{16}$$

and

$$\frac{de_r(t)}{dt}\frac{dr(t)}{dt} \leq -\frac{d\hat{r}(t)}{dt}\frac{dr(t)}{dt}, \tag{17}$$

if

$$\frac{d\hat{r}(t)}{dt} > \frac{dr(t)}{dt} \quad \forall t, \tag{18}$$

as stated by the hypothesis in (4). Then

$$\frac{d\hat{r}(t)}{dt}\frac{dr(t)}{dt} > \left(\frac{dr(t)}{dt}\right)^2 \quad \forall t, \tag{19}$$

and thus

$$\frac{d\hat{r}(t)}{dt}\frac{dr(t)}{dt} > 0 \quad \forall t. \tag{20}$$

Considering that

$$\mathcal{M} > 0 \quad \forall t, \tag{21}$$

as stated by the hypothesis in (5), then:

$$\frac{dV(e_r(t))}{dt} < 0 \quad \forall t. \tag{22}$$

Thus, (11) is uniformly asymptotically stable and (6) is proven. $\square$

**Proposition 2.** *The dynamic system*

$$\begin{aligned}\frac{d\eta(t)}{dt} &= -k_{app}\eta(t) - k_{app}^2 y(t) \\ \hat{r}(t) &= \eta(t) + k_{app}y(t),\end{aligned} \tag{23}$$

*where function $\eta(t) \in \mathbb{R}$, solves the problem defined in Problem 1.*

A supplementary variable is defined as:

$$\eta(t) = \hat{r}(t) - \mathcal{N}(y(t)), \tag{24}$$

where $\mathcal{N}(y(t))$ is a function to be designed with $\mathcal{N}(y(t)) \in \mathbb{R}$. Let

$$\mathcal{M}\frac{dy(t)}{dt} = \frac{dN(y(t))}{dt} = \frac{dN(y(t))}{dy(t)}\frac{dy(t)}{dt}. \tag{25}$$

If $\mathcal{N}(y(t)) = k_{app}y(t)$, then $\mathcal{M} = k_{app}$. Then the asymptotical stability is always guaranteed for $k_{app} > 0$ and the rate of convergence can also be specified by $k_{app} > 0$. From (24), the second part of (23) follows:

$$\hat{r}(t) = \eta(t) + k_{app}y(t). \tag{26}$$

Differentiating (26), it follows that

$$\frac{d\eta(t)}{dt} = \frac{d\hat{r}(t)}{dt} - k_{app}\frac{dy(t)}{dt}. \tag{27}$$

Considering (27) with $\frac{d\hat{r}(t)}{dt} = -k_{\text{app}}\left(\hat{r}(t) - \frac{dy(t)}{dt}\right)$ from (7) with $\mathcal{M} = k_{\text{app}}$ combined with (26), the first part of (23) follows:

$$\frac{d\eta(t)}{dt} = -k_{\text{app}}\eta(t) - k_{\text{app}}^2 y(t). \tag{28}$$

If (23) is transformed by forward Euler, the following expression is obtained:

$$\begin{aligned}
\eta(k) &= (1 - t_s k_{\text{app}})\eta(k-1) - t_s k_{\text{app}}^2 y(k) \\
\hat{r}(k) &= \eta(k) + k_{\text{app}}y(k),
\end{aligned} \tag{29}$$

and is a discrete differential equation, where $t_s$ indicates the sampling time with $t \in \mathbb{R}$, and $\hat{r}(k)$, $\eta(k)$, and $y(k)$ are discrete variables with $k \in \mathbb{N}$.

Figure 1 presents a graphical representation of the proposed algorithm structure with the discrete input signal $y(k)$, the recursive calculator for the parameter $a_2$ of the linear least squares method (LSM) and the differential estimator with the discrete approximated derivative function $\hat{r}(k)$.

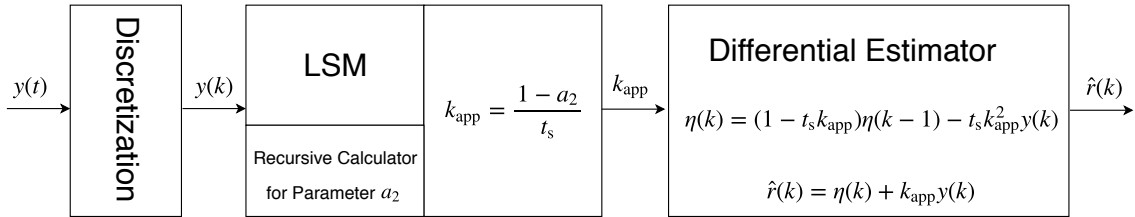

**Figure 1.** Graphical representation of the algorithm structure and its components. LSM: linear least squares method

Transforming the equations represented by (29) with $\mathcal{Z}$-transform, the following forms are obtained:

$$\begin{aligned}
H(z) &= -t_s k_{\text{app}}^2 z^{-1}Y(z) + \left(1 - t_s k_{\text{app}}\right)z^{-1}H(z), \\
\hat{R}(z) &= H(z) + k_{\text{app}}Y(z),
\end{aligned} \tag{30}$$

which yields to

$$\hat{R}(z) = \frac{k_{\text{app}}\left(1 - z^{-1}\right)Y(z)}{1 - \left(1 - t_s k_{\text{app}}\right)z^{-1}}. \tag{31}$$

As described earlier, the objective of a minimum variance approach is to minimize the variation of an output of a system with respect to a desired output signal in the presence of noise. The following theorem gives a result to determine a suboptimal $k_{\text{app}}$ to achieve a defined suboptimality.

**Theorem 1.** *Considering*

$$e_r(t) = r(t) - \hat{r}(t), \tag{32}$$

*and according to the forward Euler discretization by $\mathcal{Z}$-transform of (1), it follows that*

$$R(z) = \frac{z^{-1} - 1}{t_s}Y(z). \tag{33}$$

*Then it is possible to find a unique value of parameter $k_{\text{app}} = \frac{1-a_2}{t_s}$ of (31), which guarantees a* suboptimal *minimum of $e_r(t)$ at each $k$, where $a_2$ is a parameter to be calculated recursively using the linear least squares method (LSM).*

**Proof.** Assuming the following model:

$$e_r(k) = a_1 e_r(k-1) + a_2 e_r(k-2) + b_1 r(k-1) + b_2 r(k-2) + n(k) + c_1 n(k-1) + c_2 n(k-2), \tag{34}$$

where coefficients $a_1$, $a_2$, $b_1$, $b_2$, and $c_1$, $c_2$ belong to $\mathbb{R}$ and need to be estimated, $n(k)$ denotes the white noise. At the next sample, (34) becomes:

$$e_r(k+1) = a_1 e_r(k) + a_2 e_r(k-1) + b_1 r(k) + b_2 r(k-1) + n(k+1) + c_1 n(k) + c_2 n(k-1). \quad (35)$$

The prediction at time $k$ is:

$$\hat{e}_r(k+1/k) = a_1 e_r(k) + a_2 e_r(k-1) + b_1 r(k) + b_2 r(k-1) + c_1 n(k) + c_2 n(k-1). \quad (36)$$

Considering that

$$J = E\{e_r^2(k+1/k)\} = E\{[\hat{e}_r(k+1/k) + n(k+1)]^2\}$$

and assuming that the noise is not correlated to signal $e_r(k)$, it follows that

$$E\{[\hat{e}_r(k+1/k) + n(k+1)]^2\} = E\{[\hat{e}_r(k+1/k)]^2\} + E\{[n(k+1)]^2\} = E\{[\hat{e}_r(k+1/k)]^2\} + \sigma_n^2, \quad (37)$$

where $\sigma_n^2$ is defined as the variance of the white noise. The goal is to find $\hat{r}(k)$ such that:

$$\hat{e}_r(k+1/k) = 0. \quad (38)$$

It is possible to write (34) as:

$$n(k) = e_r(k) - a_1 e_r(k-1) - a_2 e_r(k-2) - b_1 r(k-1) - b_2 r(k-2) - c_1 n(k-1) - c_2 n(k-2). \quad (39)$$

Considering the effect of the noise as follows:

$$c_1 n(k-1) + c_2 n(k-2) \approx c_1 n(k-1), \quad (40)$$

and transforming (39) using $\mathcal{Z}$-transform, then

$$N(z) = E_r(z) - a_1 z^{-1} E_r(z) - a_2 z^{-2} E_r(z) - b_1 z^{-1} R(z) - b_2 z^{-2} R(z) - c_1 z^{-1} N(z), \quad (41)$$

and

$$N(z) = \frac{(1 - a_1 z^{-1} - a_2 z^{-2}) E_r(z)}{1 + c_1 z^{-1}} \frac{-(b_1 z^{-1} + b_2 z^{-2}) R(z)}{1 + c_1 z^{-1}}, \quad (42)$$

where $z \in \mathbb{C}$ and represents the well-known complex variable. The approximation in (40) is equivalent to considering the following assumption:

$$\|c_2\| << \|c_1\|. \quad (43)$$

In other words, the assumption stated in (43) means that the noise model of (39) is assumed to be a model of the first order. Considering the $\mathcal{Z}$-transform of (36) with $c_1 n(k-1) + c_2 n(k-2) \approx c_1 n(k-1)$, then

$$z\hat{E}_r(z) = a_1 E_r(z) + a_2 z^{-1} E_r(z) + b_1 R(z) + b_2 z^{-1} R(z) + c_1 N(z). \quad (44)$$

Considering that

$$N(z) = \frac{(1 - a_1 z^{-1} - a_2 z^{-2}) E_r(z)}{1 + c_1 z^{-1}} \frac{-(b_1 z^{-1} + b_2 z^{-2}) R(z)}{1 + c_1 z^{-1}} \quad (45)$$

and inserting (45) into (44), it follows that

$$z\hat{E}_r(z) = a_1 E_r(z) + a_2 z^{-1} E_r(z) + b_1 R(z) + b_2 z^{-1} R(z)$$
$$+ c_1 \left( \frac{(1 - a_1 z^{-1} - a_2 z^{-2}) E_r(z)}{1 + c_1 z^{-1}} \frac{-(b_1 z^{-1} + b_2 z^{-2}) R(z)}{1 + c_1 z^{-1}} \right). \quad (46)$$

According to (38), then $\hat{E}_r(z) = 0$, and through some calculations the following expression is obtained:

$$c_1 E_r(z) = a_1 E_r(z) + c_1 a_1 z^{-1} E_r(z) + a_2 z^{-1} E_r(z) + c_1 a_2 z^{-2} E_r(z) + b_1 R(z) + c_1 b_1 z^{-1} R(z)$$
$$+ b_2 z^{-1} R(z) + c_1 b_2 z^{-2} R(z) + c_1 (1 - a_1 z^{-1} - a_2 z^{-2}) E_r(z) - c_1 (b_1 z^{-1} + b_2 z^{-2}) R(z). \quad (47)$$

From (47), it follows that

$$R(z) = -\frac{(a_1 + c_1 + a_2 z^{-1}) E_r(z)}{b_1 + b_2}. \quad (48)$$

Considering that

$$E_r(z) = R(z) - \hat{R}(z), \quad (49)$$

then relation (48) becomes

$$R(z) = -\frac{(a_1 + c_1 + a_2 z^{-1})(R(z) - \hat{R}(z))}{b_1 + b_2}, \quad (50)$$

and thus the derivative approximation according to the forward method is

$$R(z) = \frac{z^{-1} - 1}{t_s} Y(z). \quad (51)$$

It follows that

$$-\frac{1 - z^{-1}}{t_s} Y(z) = -\frac{(a_1 + c_1 + a_2 z^{-1})\left((1 - z^{-1}) t_s^{-1} Y(z) - \hat{R}(z)\right)}{b_1 + b_2}, \quad (52)$$

thus

$$-(1 - z^{-1})\left((b_1 + b_2) t_s^{-1} + (a_1 + c_1 + a_2 z^{-1}) t_s^{-1}\right) Y(z) = (a_1 + c_1 + a_2 z^{-1}) \hat{R}(z), \quad (53)$$

and finally,

$$\hat{R}(z) = -(1 - z^{-1}) \frac{\left((b_1 + b_2) t_s^{-1} + (a_1 + c_1 + a_2 z^{-1}) t_s^{-1}\right) Y(z)}{a_1 + c_1 + a_2 z^{-1}}, \quad (54)$$

which can be written as

$$\hat{R}(z) = (1 - z^{-1}) \frac{\left((b_1 + b_2) t_s^{-1} + (a_1 + c_1 + a_2 z^{-1}) t_s^{-1}\right) Y(z)}{-a_1 - c_1 - a_2 z^{-1}}. \quad (55)$$

Recalling (31),

$$\hat{R}(z) = \frac{k_{app} \left(1 - z^{-1}\right) Y(z)}{1 - \left(1 - t_s k_{app}\right) z^{-1}}, \quad (56)$$

and comparing (54) with (56), the denominator constraints are

$$-c_1 - a_1 = 0 \quad (57)$$

and

$$a_2 = \left(1 - t_s k_{app}\right), \quad (58)$$

together with

$$k_{\text{app}} = \left( \frac{1 - a_2}{t_{\text{s}}} \right). \tag{59}$$

Parameter $a_2$ can be calculated by LSM.

The numerator constraint is the following:

$$(b_1 + b_2)t_{\text{s}}^{-1} + (a_1 + c_1 + a_2 z^{-1})t_{\text{s}}^{-1} = k_{\text{app}}.$$

Considering the conditions of the denominator, we obtain

$$(b_1 + b_2)t_{\text{s}}^{-1} + \left(-1 + (1 - t_{\text{s}}k_{\text{app}}) z^{-1}\right)t_{\text{s}}^{-1} = k_{\text{app}}.$$

This yields

$$b_1 + b_2 = k_{\text{app}}t_{\text{s}} + 1 - (1 - t_{\text{s}}k_{\text{app}}) z^{-1}.$$

$k_{\text{app}}$ being, in our context, a function of time $k_{\text{app}} = k_{\text{app}}(t)$, it is possible to write in $\mathcal{Z}$-domain as follows:

$$b_1(z) + b_2(z) = k_{\text{app}}(z)t_{\text{s}} + 1 - (1 - t_{\text{s}}k_{\text{app}}(z)) z^{-1},$$

and thus consider the back $\mathcal{Z}$-transform

$$b_1(k) + b_2(k) = k_{\text{app}}(k)t_{\text{s}} + 1 - (1 - t_{\text{s}}k_{\text{app}}(k-1)).$$

If $t_{\text{s}}$ is small enough, $k_{\text{app}}(k) \approx k_{\text{app}}(k-1)$. This implies

$$b_1(k) + b_2(k) = 0. \tag{60}$$

□

**Remark 1.** *Conditions (57), (59), and (60) guarantee that signal $\hat{r}(t)$ equals $y(t)$. Nevertheless, obtaining the rejection of the noise coefficients $b_1$, $b_2$, and $c_1$ should be adaptively calculated using LSM. In our tests, in order to reduce the calculation load, the following conditions are considered: $c_1 = b_1 = b_2 = 0$.*

## 3. Using an Adaptive Kalman Filter to Approximate a Derivative Controller

Assuming that the polynomial that approximates the derivative of the signal is of the first order as follows:

$$\hat{y}(t) = p_0 + p_1 t, \tag{61}$$

in which $\hat{y}(t)$ represents the polynomial approximation of the signal $y(t)$. The following adaptive Kalman filter (KF) can be implemented in which at each sampling time, constant parameters $p_0$ and $p_1$ should be calculated. The following state representation is obtained:

$$\begin{bmatrix} \dot{\hat{y}}(t) \\ \dot{\hat{r}}(t) \end{bmatrix} = \begin{bmatrix} 0 & 1 \\ 0 & 0 \end{bmatrix} \begin{bmatrix} \hat{y}(t) \\ \hat{r}(t) \end{bmatrix}. \tag{62}$$

It should be noted that $\dot{\hat{y}}(t) = \hat{r}(t)$ represents the approximation of the derivative of the signal as proposed in Proposition 1. In this sense, according to the following general notation, $\mathbf{x}(t) \in \mathbb{R}^2$, $u(t) \in \mathbb{R}$, and $y(t) \in \mathbb{R}$, then:

$$\begin{cases} \dot{\mathbf{x}}(t) = \mathbf{A}\mathbf{x}(t) \\ y(t) = \mathbf{H}\mathbf{x}(t), \end{cases} \tag{63}$$

where

$$\dot{\mathbf{x}}(t) = \begin{bmatrix} \dot{\hat{y}}(t) \\ \dot{\hat{r}}(t) \end{bmatrix} \text{ and } \mathbf{A} = \begin{bmatrix} 0 & 1 \\ 0 & 0 \end{bmatrix}, \tag{64}$$

and $\mathbf{H} = \begin{bmatrix} 1 & 0 \end{bmatrix}$. Considering the discretization of system (63), the following discrete system is obtained:

$$\begin{cases} \dot{\mathbf{x}}(k/k-1) = \mathbf{A}_d \mathbf{x}(k-1/k-1) + \mathbf{Q}_w \\ y(k/k) = \mathbf{H}\mathbf{x}(k/k) + \zeta, \end{cases} \tag{65}$$

where $\mathbf{Q}_w$ is the process noise covariance matrix, and $\zeta$ is the measurement noise covariance. The discrete forms of matrix $\mathbf{A}$ of (64) are represented by $\mathbf{A}_d$, respectively. If the forward Euler method with the sampling time $t_s$ is applied, the following matrices are obtained:

$$\mathbf{A}_d = \begin{bmatrix} 1 & t_s \\ 0 & 1 \end{bmatrix}. \tag{66}$$

The a priori predicted state is

$$\mathbf{x}(k/k-1) = \mathbf{A}_d \mathbf{x}(k-1/k-1), \tag{67}$$

and the a priori predicted covariance matrix is

$$\mathbf{P}(k/k-1) = \mathbf{A}_d \mathbf{P}(k-1/k-1)\mathbf{A}_d^{\mathrm{T}} + \mathbf{Q}_w. \tag{68}$$

The following equations state the correction (a posteriori prediction) of the adaptive KF:

$$\begin{aligned} \mathbf{K}(k) &= \mathbf{P}(k/k-1)\mathbf{H}^{\mathrm{T}}\left(\mathbf{H}\mathbf{P}(k/k-1)\mathbf{H}^{\mathrm{T}} + \zeta\right)^{-1}, \\ \mathbf{x}(k/k) &= \mathbf{x}(k/k-1) + \mathbf{K}(k)\left(y(k) - \mathbf{H}\mathbf{x}(k/k-1)\right), \\ \mathbf{P}(k/k) &= \mathbf{P}(k/k-1) - \mathbf{K}(k)\mathbf{H}\mathbf{P}(k/k-1), \end{aligned} \tag{69}$$

where $\mathbf{K}(k)$ is the Kalman gain.

**Remark 2.** *It should be noted that matrix $\mathbf{Q}_w$ (process covariance noise) consists of the following structure:*

$$\mathbf{Q}_w = \begin{bmatrix} 0 & 0 \\ 0 & R_{2,2} \end{bmatrix}, \tag{70}$$

*in which $R_{2,2}$ states a squared variance. The reason for this structure is that the first equation corresponding to matrix $\mathbf{A}$ of (64) is a definition of the velocity, and in this sense, no uncertainty should be set. The second equation is an equation which considers a comfortable condition, but of course it is not true. In this case, an uncertainty variable must be set. According to our experience, a very wide system needs a very wide uncertainty variable to be set.*

**Remark 3.** *Parameter $R_{2,2}$ is adapted by directly using the definition of the covariance as follows:*

$$R_{2,2} = (e_a(N) - \text{mean}(e_a(1:N)))^2,$$

*in which a mean value with the last N values of error $e_a(k)$ are calculated.*

## 4. Results and Discussion

The following section presents the results using a derivative realized through the adaptive KF and the proposed algorithm. The results are compared with the exact mathematical derivative.

In order to reduce the calculation load in the case study, the following conditions were considered: $c_1 = b_1 = b_2 = 0$. This approach allows us to also compare the results with a polynomial adaptive KF method.

Figure 2 shows an ideal measured position $y(t)$ signal. Therefore neither noise nor faults are present in the measured data. Figure 3 shows the approximated derivative of the measured sine function, and the result of this is shown in detail in Figure 4. With this graphical representation of the result, it is visible that the adaptive KF, compared with the proposed algorithm structure, shows a better performance.

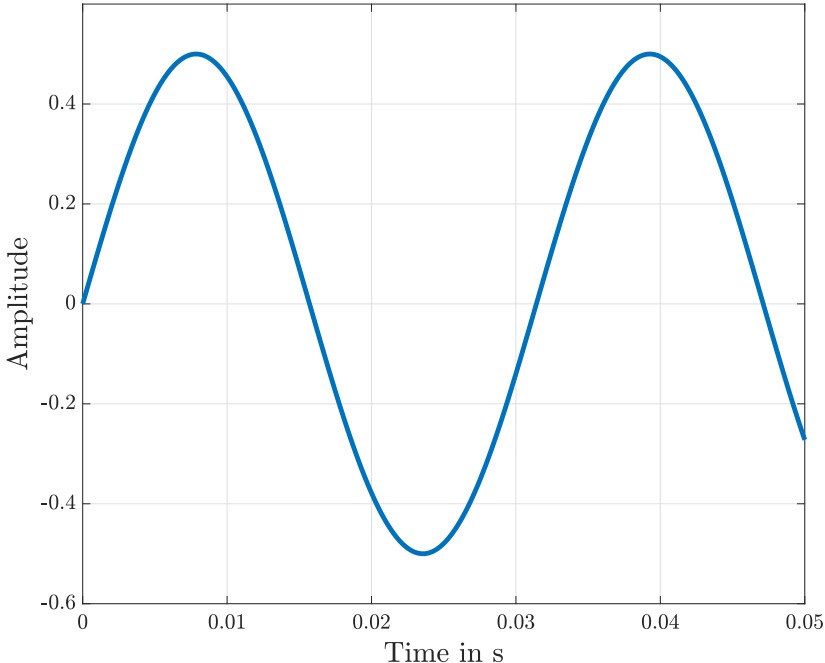

**Figure 2.** Graphical representation of the position $y(t)$.

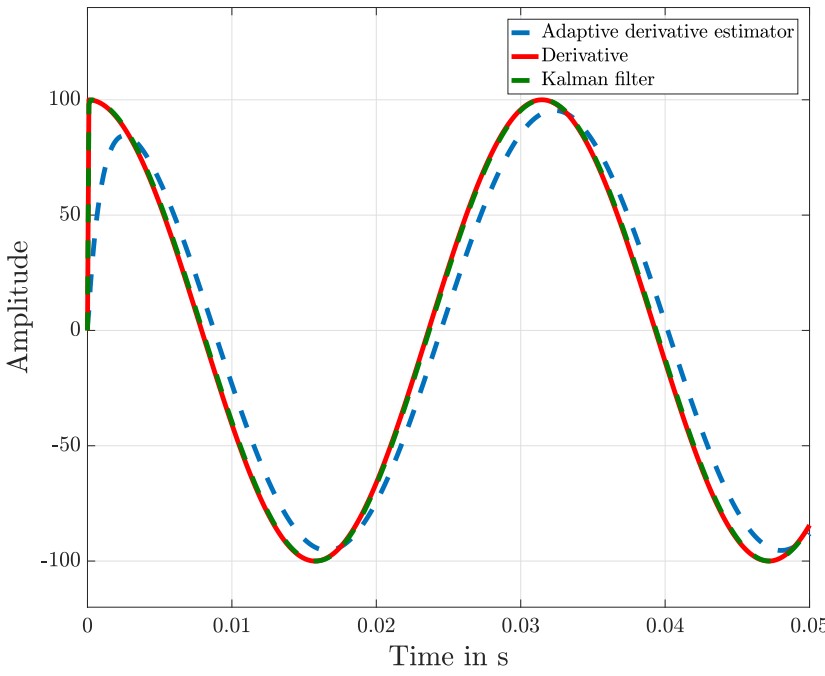

**Figure 3.** Graphical representation of the resulting velocity and its estimations from Figure 2.

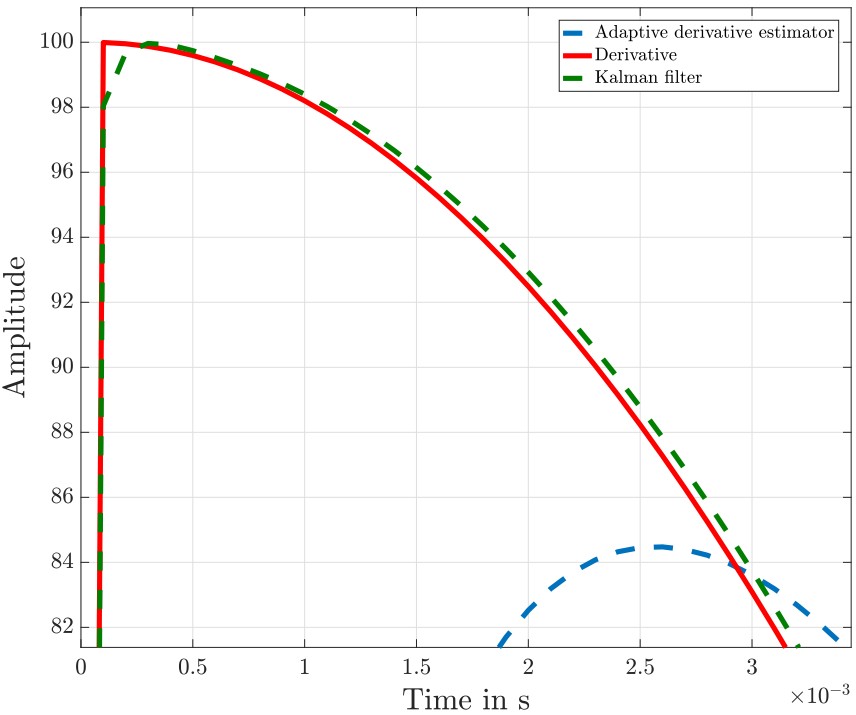

**Figure 4.** Detailed representation of Figure 3 and its estimations.

Figure 5 shows the position signal used, $y_{\text{noise}}(t)$, in which noise in superposition is added, and the graphical representation of the resulting velocity is shown in Figure 6. The signal used, $y_{\text{noise}}(t)$, is represented in Figure 7 together with a fault in its measurement. Figure 8 shows the approximation derivative of the measured signal, and details of this result are shown in Figure 9. Also in the case of the presence of faults and using a more appropriate adaption of the adaptive KF, the two methods offer similar results.

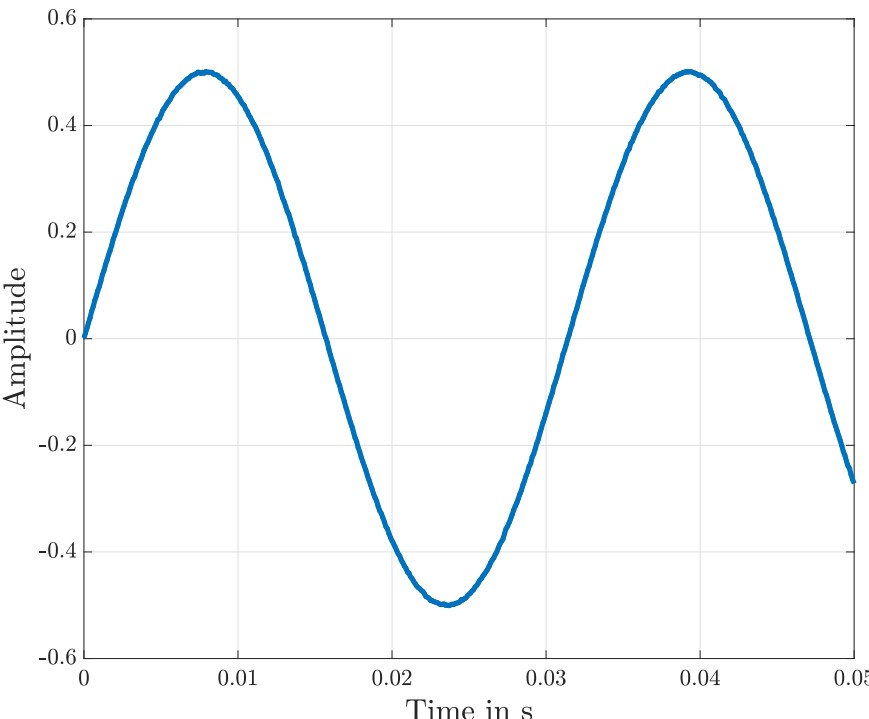

**Figure 5.** Graphical representation of the position $y(t)$ with noise.

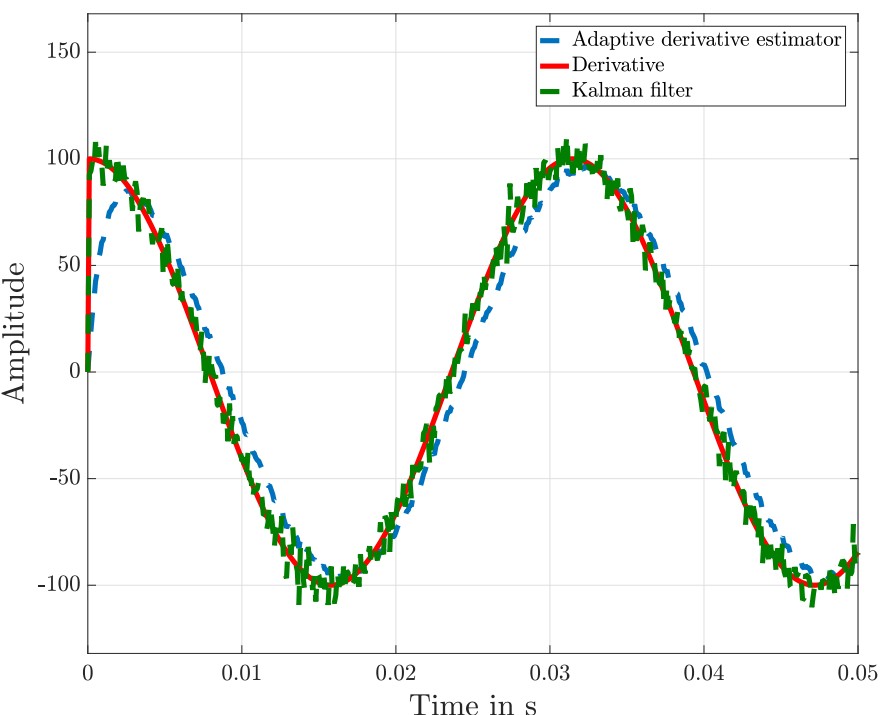

**Figure 6.** Detailed representation of Figure 5 and its estimations.

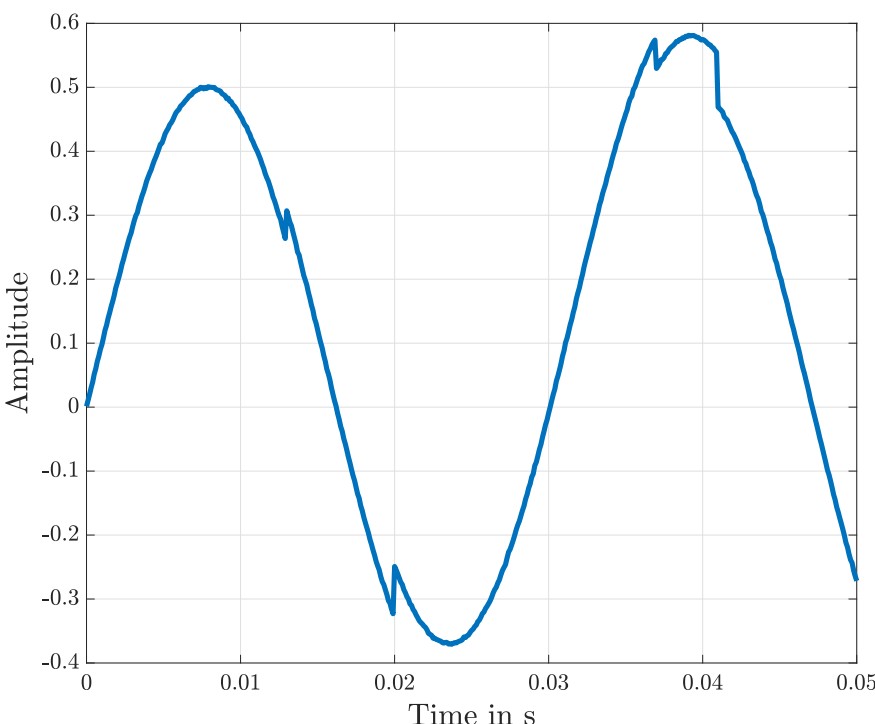

**Figure 7.** Graphical representation of the position $y(t)$ with noise and faults.

Table 1 presents the results of different input signals and the estimation errors of both adaptive estimators based on the Euclidean norm.

The adaptive derivative estimator shows worse results with respect to the results obtained by the derivative obtained through the adaptive KF in the case of missing noise and faults.

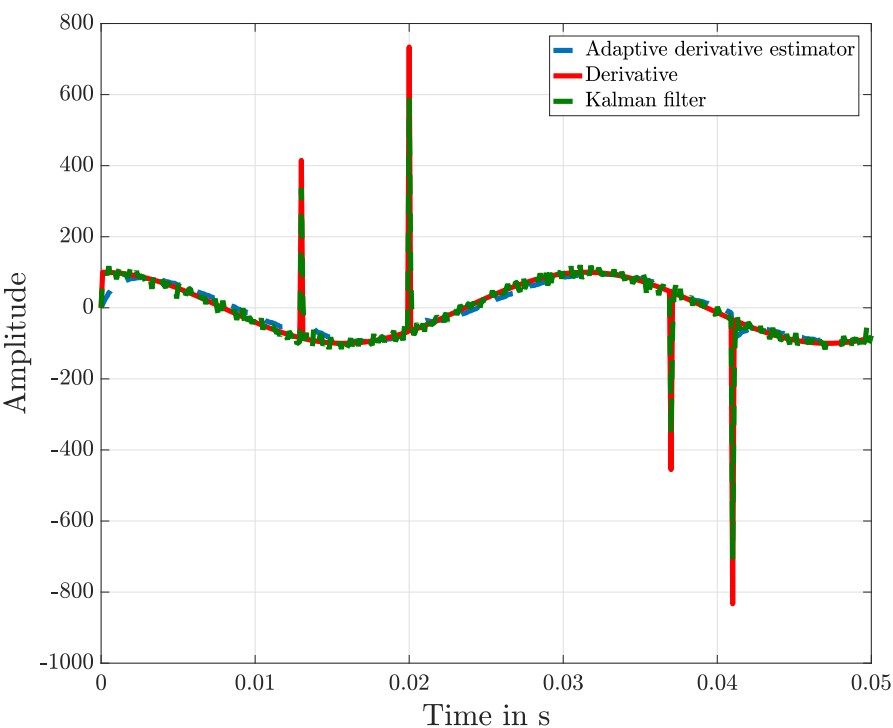

**Figure 8.** Graphical representation of the resulting velocity from Figure 7 and its estimations with noise and faults.

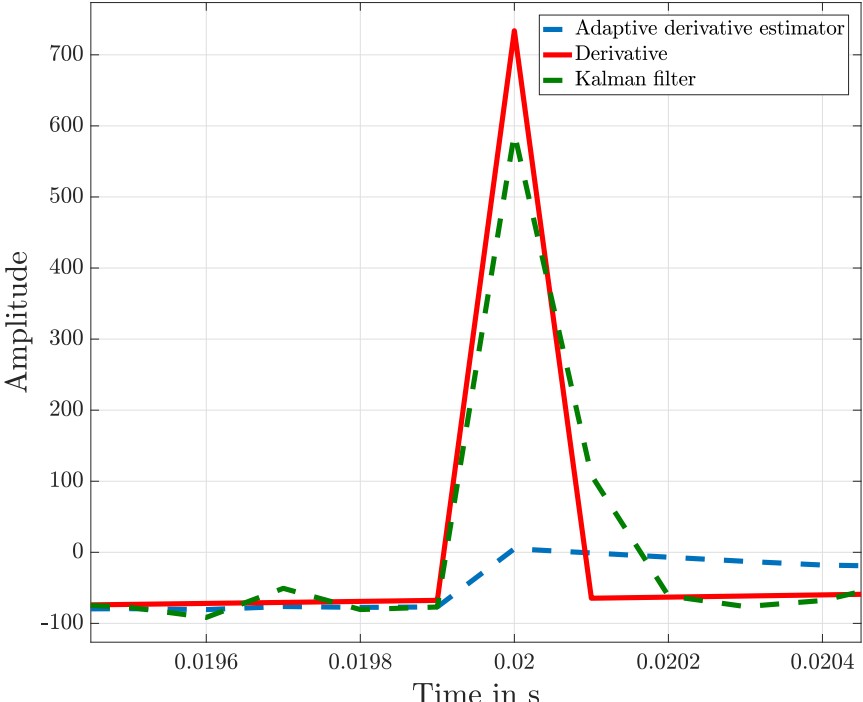

**Figure 9.** Detailed representation of Figure 7 with its estimations.

In case of the presence of noise and using a more appropriate adaption of the adaptive KF, the two methods offer similar results. The proposed algorithm structure shows better results in the presence of faults.

Figure 10 shows that a part of a chirp input signal starts from a frequency of 200 Hz and in 0.3 s reaches to 2 kHz. Because of the presence of the derivative, the signal to be approximated changes its

amplitude linearly as a function of the frequency over time. The low-frequency part of the chirp input signal is presented in Figure 11, and the high-frequency part is shown in Figure 12. It also shows the resulting tracking of the polynomial adaptive KF and the proposed algorithm structure.

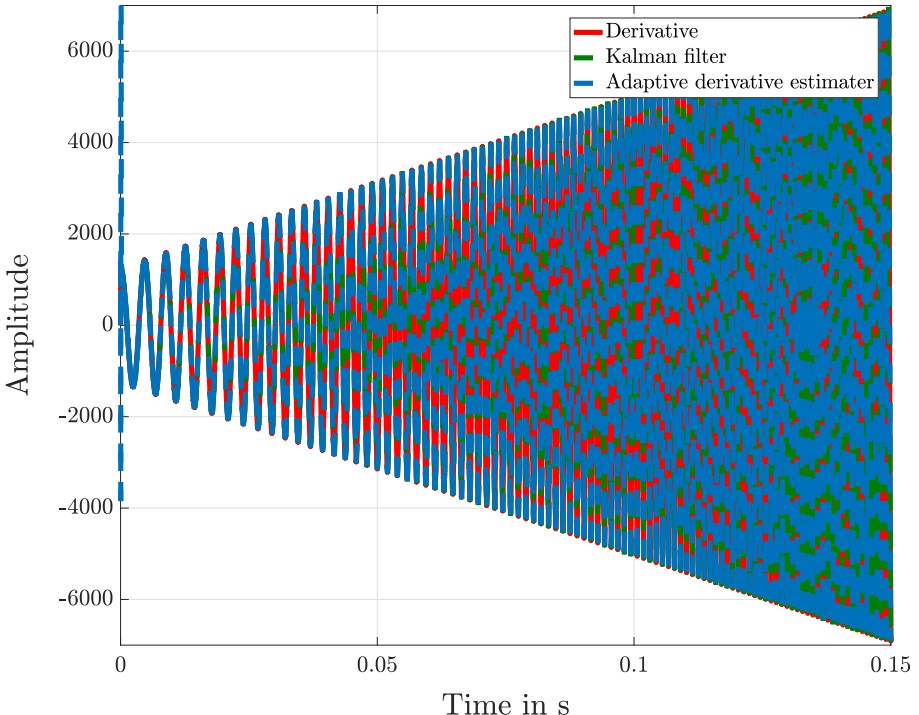

**Figure 10.** Graphical representation of chirp input signal with a starting frequency of 200 Hz.

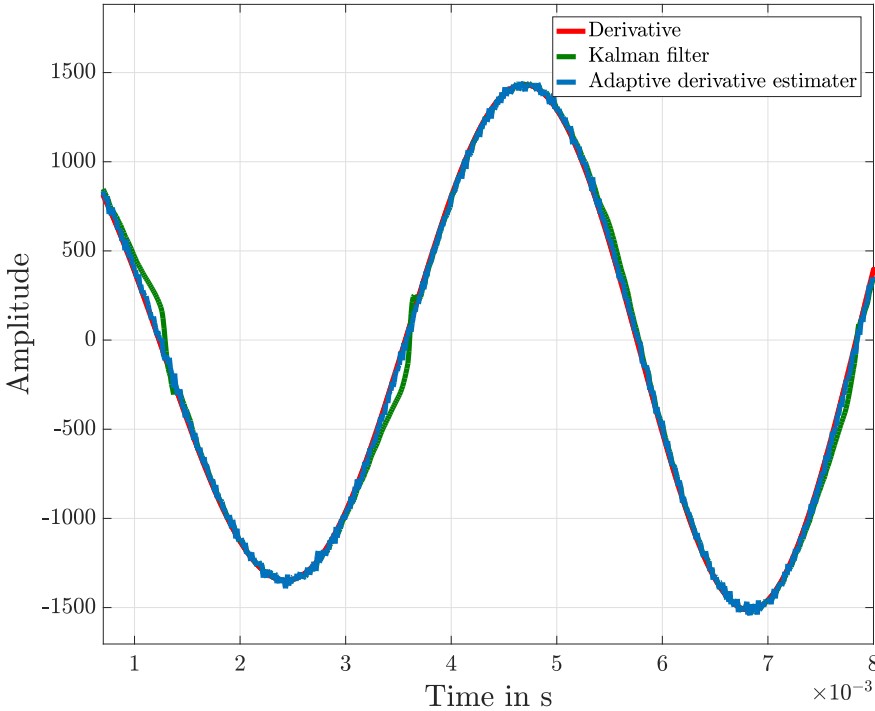

**Figure 11.** Detailed representation of the low-frequency part of the chirp input signal and its estimations from Figure 10.

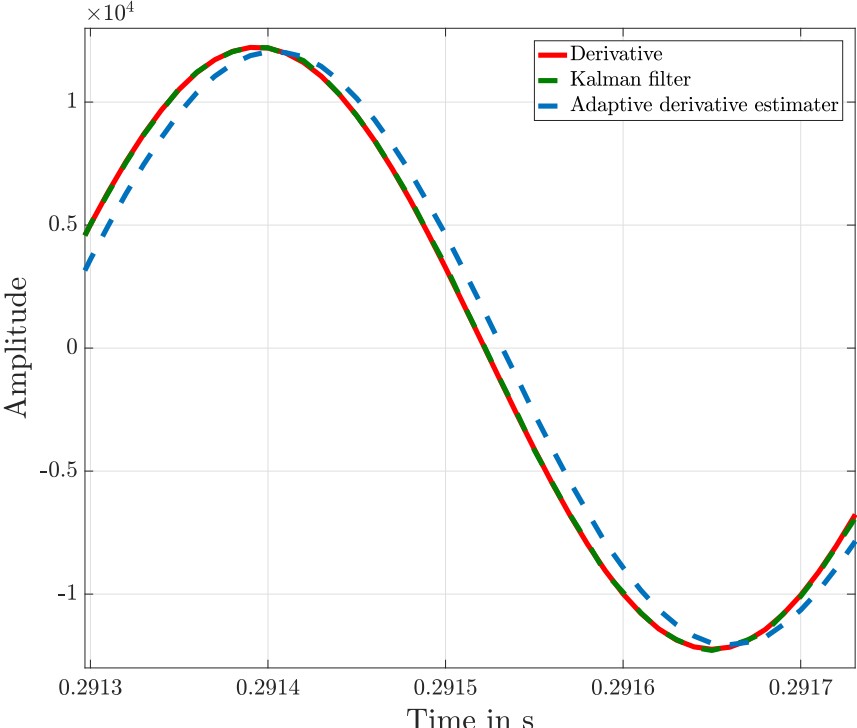

**Figure 12.** Detailed representation of the high-frequency part of the chirp input signal and its estimations from Figure 10.

**Table 1.** Overview of the estimation errors of different input signals $y(t)$ using the Euclidean norm.

| Input Signal | Adaptive Kalman Filter | Adaptive Derivative Estimator |
|:---:|:---:|:---:|
| $\sin^2(t)$ | $4.04 \times 10^{-3}$ | 13.79 |
| $\sin^2(t)$ with noise | 12.06 | 13.95 |
| $\sin^2(t)$ with noise & $\sin(t)$ | 32.34 | 32.12 |
| $\sin^2(t)$ with noise & $\sin(t)$ & fault | 159.0 | 34.73 |
| $\sin^3(t)$ | $1.4 \times 10^{-3}$ | 4.29 |
| $\sin^3(t)$ with noise | 12.05 | 4.41 |
| $\sin^3(t)$ with noise & $\sin(t)$ | 32.34 | 22.98 |
| $\sin^3(t)$ with noise & $\sin(t)$ & fault | 159.0 | 27.10 |

A fault is defined as an abrupt change in the amplitude over time, and in this sense is characterized by a high amplitude and high frequencies. In this context, it is clear that the proposed algorithm can better localize the faults because it is tuned through the choice of $t_s$ on the desired signal.

The proposed algorithm structure does not need to be tuned or, in other words, is tuned using the sample time, which in general is fixed in terms of upper bound by the Shannon Theorem. Now $k_{app}$ represents the time constant of our approximating derivative, which is calculated adaptively through the least squares method. The simulation shows that once the KF is tuned, this shows better results at high frequencies with respect to the algorithm structure, but in the range of the frequency in which $t_s$ is consistently chosen, the proposed algorithm shows similar results as those offered by KF.

## 5. Conclusions

This paper deals with an approximation of a first derivative of a input signal using a dynamic system of the first order to avoid spikes and noise. It presents an adaptive derivative estimator for fault-detection using a suboptimal dynamic system in detail. After formulating the problem, a proposition and a theorem were proven for a possible approximation structure, which consists

of a dynamic system. In particular, a proposition based on a Lyapunov approach was proven to show the convergence of the approximation. The proven theorem is constructive and directly shows the suboptimality condition in the presence of noise. A comparison of simulation results with the derivative realized using an adaptive KF and with the exact mathematical derivative were presented. It was shown that the proposed adaptive suboptimal auto-tuning algorithm structure does not depend on the setting of the parameters. Based on these results, an adaptive algorithm was conceived to calculate the derivative of an input signal with convergence in infinite time. The proposed algorithm showed worse results with respect to the results obtained by the derivative obtained through the adaptive KF in the case without noise and faults. In case of the presence of noise and using a more appropriate version of the adaptive KF, the two methods offer similar results. In the presence of faults, the proposed algorithm structure showed better results.

**Author Contributions:** Conceptualization, M.S. and P.M.; software, M.S. and P.M.; validation, M.S. and P.M.; formal analysis, M.S. and P.M.; investigation, M.S. and P.M.; resources, M.S. and P.M.; data curation, M.S. and P.M.; writing–original draft preparation, M.S. and P.M.; writing–review and editing, M.S. and P.M.; visualization, M.S. and P.M.

**Funding:** This research received no external funding.

**Conflicts of Interest:** The authors declare no conflict of interest.

## Nomenclature

| | |
|---|---|
| $a_2$ | Adaptive least square parameter |
| $\mathbf{x}(t)$ | State vector of Kalman filter |
| $e_r(t)$ | Derivative error |
| $k$ | Discrete variable |
| $\mathbf{K}(k)$ | Kalman gain |
| $k_{\mathrm{app}}$ | Parameter |
| $n(t)$ | Noise signal |
| $N(z)$ | $\mathcal{Z}$-transformed noise signal |
| $\mathbf{Q}_w$ | Process noise covariance matrix |
| $r(t)$ | Derivative function |
| $\hat{r}(t)$ | Approximated derivative function |
| $R(z)$ | $\mathcal{Z}$-transformed derivative function |
| $\hat{r}(t)$ | Approximated derivative function |
| $Y(z)$ | $\mathcal{Z}$-transformed signal to be derivated with or without noise |
| $y(t)$ | Signal to be derivated |
| $\hat{y}(t)$ | Polynomial expression of the signal |
| $y_{\mathrm{noise}}(t)$ | Position signal with noise |
| $t_{\mathrm{s}}$ | Sampling time |
| $\zeta$ | Measurement noise covariance matrix |

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
