# Peer review of "An Adaptive Derivative Estimator for Fault-Detection Using a Dynamic System with a Suboptimal Parameter"

_algorithms, doi:10.3390/a12050101_

Round 1

Reviewer 1 Report

 I found an interesting manuscript, however, since it  claims the derivative response of a dynamic system It is necessary to bound the sort of system that are following in terms of physical condition of the fault. In my opinion the proposal has an interesting value for the community, nevertheless, it is quite blurred in terms of the observed behavior. I would suggest to spend some pages to clarify the sort of scale/frequency response of the system in order to have a clear idea of the response from the system. Similar argument would be interesting in terms of the nature of the fault since it is not possible to let this behavior as an unclear issue.

Reviewer 2 Report

The paper introduces an adaptive derivative estimator algorithm, which was designed for first derivative estimation of a measured signal.

The results presented by the Authors are not convincing. Detailed discussion of the results is missing. Each figure/table in Sect. 4 is just mentioned in one sentence. After reading the paper, there are doubts if the proposed algorithm gives better results than those based on the Kalman filter. For instance, in case of Figs. 2 and 3 the Authors have stated that " From this graphical result representation it is visible how the approximated derivative obtained with the proposed algorithm shows better results with respect to the results obtained by the derivative obtained through the adaptive Kalman filter. " However, the charts in Fig. 2 and Fig.3 show something different. In these charts the approximation obtained from Kalman filter is significantly closer to the derivative than the approximation found by the proposed algorithm. Moreover, different line types could be used to improve the clarity of the charts. According to the information presented in the manuscript, the proposed algorithm is not better than the state-of-the-art adaptive Kalman. The discussion in Sect. 4 has to be significantly extended.

The proposed algorithm should be summarized in a form of pseudo code.

It is not clear how the values in Tab. 1 were calculated.

Section 5 has to be rewritten. The second paragraph in Sect. 5 is just a copy of the abstract.

Last part of Sect. 1 includes a description of structure of the paper. In this fragment only sections 2 and 5 are mentioned. The remaining sections should be also taken into account.

There are many language errors and awkward sentences, e. g.:

line 16 " In applications are signals corrupted by measurement- and process noise and a filtering procedure is to implemented." ,

page 8 " In the presents of noise ", " It is to notice that ",

line 90 "In this section results using a derivative realized through the adaptive Kalman filter and the proposed algorithm are presented",

line 113 " the proposed algorithm does not depends on ".

Round 2

Reviewer 1 Report

I  agree with this new version of the manuscript 

Reviewer 2 Report

 The manuscript has been improved and now can be accepted publication in Algorithms.